

# Online e-learning during the COVID-19 lockdown in Trinidad and Tobago: prevalence and associated factors with ocular complaints among schoolchildren aged 11–19 years

Kingsley Ekemiri[1,2], Ngozika Ezinne[1,2], Khadeejah Kamalodeen[2], Keomi Pierre[2], Brandon Lalla[2], Onyekachukwu Amiebenomo[3,4], Diane van Staden[1], Ferial Zeried[5], Chioma Ekemiri[6], Kingsley E. Agho[7,8,9] and Uchechukwu Levi Osuagwu[8,9]

[1] Department of Optometry, Faculty of Health Sciences, University of Kwazulu-Natal, Kwazulu-Natal, Kwazulu Natal, South Africa
[2] Optometry Unit, Department of Clinical Surgical Sciences, Faculty of Medical Sciences, University of the West Indies St. Augustine, St Augustine, Trinidad and Tobago
[3] Department of Optometry, Faculty of Life Sciences, University of Benin, Benin City, Edo State, Nigeria
[4] School of Optometry and Vision Sciences, College of Biomedical Sciences, Cardiff University, Cardiff, United Kingdom
[5] Department of Optometry & Vision Sciences College of Applied Medical Sciences, King Saud University, Riyadh, Ar Riyadh, Saudi Arabia
[6] Department of Health Promotion, The University of the West Indies, St. Augustine Campus, Trinidad and Tobago
[7] School of Health Sciences, Western Sydney University, Campbelltown, NSW, Australia
[8] African Vision Research Institute, Department of Optometry, Faculty of Health Sciences, University of Natal, Durban, Kwazulu-Natal, Durban, South Africa
[9] Translational Health Research Institute (THRI), School of Medicine, Western Sydney University, Campbelltown, NSW, Australia

Corresponding author
Uchechukwu Levi Osuagwu,
l.osuagwu@westernsydney.edu.au

## ABSTRACT

**Background:** The increase in online learning during the pandemic has been linked to various ocular complaints. This study determined the prevalence and factors associated with ocular complaints among schoolchildren aged 12–19 years during the COVID-19 lockdown in Trinidad and Tobago (T&T).

**Methods:** A cross-sectional study was conducted between January and May 2021, during the COVID-19 lockdown in T&T among secondary school students studying remotely. A two-stage cluster sampling method was employed. A modified web-based Computer Vision Syndrome questionnaire was administered to students. Data on demography, duration of digital device use, and ocular complaints were collected, and multilevel logistic regression was used to determine factors associated with ocular complaints among school children, 12–19 years of age in T&T.

**Results:** A total of 435 schoolchildren (mean age, standard deviation, 15.2 ± 1.9 years range 12–19 years) responded to the questionnaire. The prevalence of self-reported symptoms of headache, blurred vision, dry eyes, itchy eyes, and double vision were 75.0%, 65.1%; 56.8%; 46.4%; and 33.5%, respectively. Schoolchildren aged 18–19 years, those that used spectacles for correction of their refractive errors, and spent more than 6 h on average on digital devices, reported a high prevalence of any ocular

complaints. Analysis also revealed that age (14–15 years) was associated with dry eyes, blurred vision, and headaches, while gender (more prevalently females) was associated with blurred vision and headache. Those that had an eye examination in the last year and schoolchildren that took action to resolve ocular complaints were more likely to experience nearly all ocular complaints.

**Conclusions:** During the COVID-19 lockdown, over three in four students in T&T reported ocular complaints from digital devices for online learning. Tailored interventional messages to reduce all forms of ocular complaints should target older students, particularly females, those who laid down when learning online *via* their devices and people who regularly examine their eyes.

# INTRODUCTION

Prior to the COVID-19 lockdown, school children in Trinidad and Tobago (T&T) learned in a typical classroom setting where students engaged in in-person, face-to-face activities, with minimal class time spent on digital devices (DD) during the school day. However, the COVID-19 pandemic and lockdown measures led to an increase in DD for various purposes, including online learning, even as people tried to foster ongoing social, business and educational engagement (*De', Pandey & Pal, 2020*). This sudden change in learning, work and communication during the pandemic became the new normal, and people needed to modify how they lived and worked (*Schieman et al., 2021*).

Globally, education has changed dramatically following the pandemic, with the phenomenal rise of e-learning, whereby teaching is undertaken remotely, on digital platforms (*Lockee, 2021*). Research suggests that online learning increases retention of information and takes less time (*Smith, Clark & Blomeyer, 2005*), and as such, the changes in our school system caused by the pandemic might be here to stay. Consequently, more students have had to depend on DD to engage with their educators and peers, leading to more time on DD (*Alabdulkader, 2021*; *Ganne et al., 2020*; *Usgaonkar, Shet Parkar & Shetty, 2021*). To maintain a clear, sharp image with the eye when working with DD, continual contraction of the eye's extraocular and ciliary muscles is required. When prolonged, the visual task may exceed the visual system's capacity (*Moulakaki et al., 2017*). When used at near working distances, extended use of digital screens such as computers, tablets, e-readers, and smartphones result in eye strain, ocular discomfort, dry eye, diplopia, and blurred vision (*Shrestha, Mohamed & Shah, 2011*), which are a group of eye and vision-related problems commonly known as digital eye strain (DES) or computer vision syndrome (CVS) (*Akagi et al., 2019*; *Mylona et al., 2020*). This problem is reported to affect up to half the video display unit (VDU) users (*Courtin et al., 2016*; *Sheppard & Wolffsohn, 2018*), with some people presenting with one or more ocular complaints as well as other systemic symptoms like headaches, neck, back and shoulder pains (*Chetty et al.,*

*2020*) which are likely to escalate with an increase in the use of devices (*Loh & Redd, 2008*; *Mohan et al., 2021*; *Usgaonkar, Shet Parkar & Shetty, 2021*). In India, a recent cross-sectional study found that compared with the background population, significantly more students who participated in online classes reported ocular complaints (*Ganne et al., 2020*), the onset of which has been linked to inaccurate contrast levels and glare on screens (*Loh & Redd, 2008*). There is a high demand for the visual system when using digital screens, which may be due to the viewing distances, posture, and angles required for working on computer screens (*Mylona et al., 2020*). In an observational cross-sectional study conducted in Saudi Arabia (*Alabdulkader, 2021*), the researcher found that the rate of self-reported ocular complaints was increased by about 78% during the pandemic, and the proportion with eye strain rose by 51% compared with the pre-lockdown period. Furthermore, *Mohan et al. (2021)* reported that itching and headache were widespread among high school children who participated in online e-learning for more than two daily hours, while the only study from a Caribbean country reported that more than 60% of the university students in Jamaica complained of eye strain and 'eye burn' following the use of DD (*Mowatt et al., 2018*).

The Republic of T&T, where the current study was undertaken, is a twin island state located at the southernmost tip of the Caribbean archipelago, with Trinidad as the largest of the islands. The country has a multi-ethnic population of approximately 1.4 million, mostly East Indians (40.3%) and (Africans 39.6%), with about 18.4% of mixed and 1.7% of other ethnic groups (*Baboolal, Davis & McRae, 2014*; *Ekemiri et al., 2021*). In T&T, a more significant percentage of the population (84%) are connected to the internet more than anywhere else in the region. The proportion with high-speed broadband and mobile internet usage increased by 66% and 64%, respectively, over the past 12 months (*Smith & Stamatakis, 2021*). Despite such high internet penetration and increased usage during the pandemic, no data exists on the impact of internet use for online learning on the visual system in the Caribbean and T&T in particular. The present study was conducted to determine the prevalence and factors associated with ocular complaints due to remote online learning among schoolchildren during the COVID-19 pandemic. The findings of this study will provide data on the prevalence of ocular complaints among school children who use DD for online learning. The data will help identify the population at greater risk of adverse outcomes from the uncontrolled use of DD targeted for improved outcomes.

# MATERIALS AND METHODS

## Ethical consideration

The study followed the Declaration of Helsinki and was approved by the University of the West Indies' Ethics Committee (CREC-SA.0712/01/202). Assent, consent and permission (where necessary) were obtained from the selected secondary school students and their parents. These were sent and returned *via* email to the parents of the selected students by the class teachers as directed by each of the participating school administrations. Before recruitment, participants were informed of the studys purpose, duration, and anonymity. Additionally, parents of the school children were informed that their child's data would be used for research purposes without disclosing their identities. The parents

and their wards had up to 5 months for consent and completion of the questionnaire. Nonconsenting students or those whose parents/guardians did not provide consent and those with ocular diseases other than nearsightedness, farsightedness, and astigmatism were excluded in this study.

### Study design

This was a cross-sectional, school-based descriptive study of school learners aged 12–19 years who attended government-owned secondary schools in T&T.

### Sample selection

T&T has 134 state-owned secondary schools, 124 (92.5%) of which are in Trinidad and are distributed across the seven districts (*Lochan & Barrow, 2008*). The study population was chosen using a two-stage cluster sampling technique. The first stage involved creating a list of all schools in each district and labeling them according to the number of schools in each district. Schools were selected using a random number generator (https://stattrek.com/statistics/random-number-generator.aspx), resulting in a simple random sampling of seven schools from Trinidad's seven districts. The second stage involved the use of simple random selection to select an average of 10 pupils from each class 1–6 in each secondary school (60 students per school). The study enrolled students from selected secondary schools in Trinidad and Tobago.

### Questionnaire design

A structured questionnaire adapted from a previous study (*Mohan et al., 2021*) was administered electronically to secondary schoolchildren studying remotely in T&T during the COVID-19 pandemic between January and May 2021. The questionnaire shown in Table S1 was created using Survey Legend's free-online survey tool (SurveyLegend AB, Malmö, Sweden) and modified to suit the study objectives. It included information about the student's demography, current study status, time spent per day using a digital device, type of digital device used, visual ergonomics, and the pattern of ocular complaints.

### Sample size determination

The sample size was based on an estimated proportion of moderate and severe vision impairment of 16% in the Caribbean from a previous study (*Burton et al., 2021*). The significance level was 5%, an allowance of 0.05 alpha error and a study power of 80%. The calculated sample size was 207. We considered the cluster sampling design effect of two based on a previous study (*Ebri, Govender & Naidoo, 2019*), yielding the minimum sample size of 414 children.

### Dependent variable

The dependent variables were five ocular complaints (blurred vision, double vision, dry eyes, headache and itchy eyes) and any ocular complaint which was used to describe the presence of any of the complaints as shown in Fig. 1.

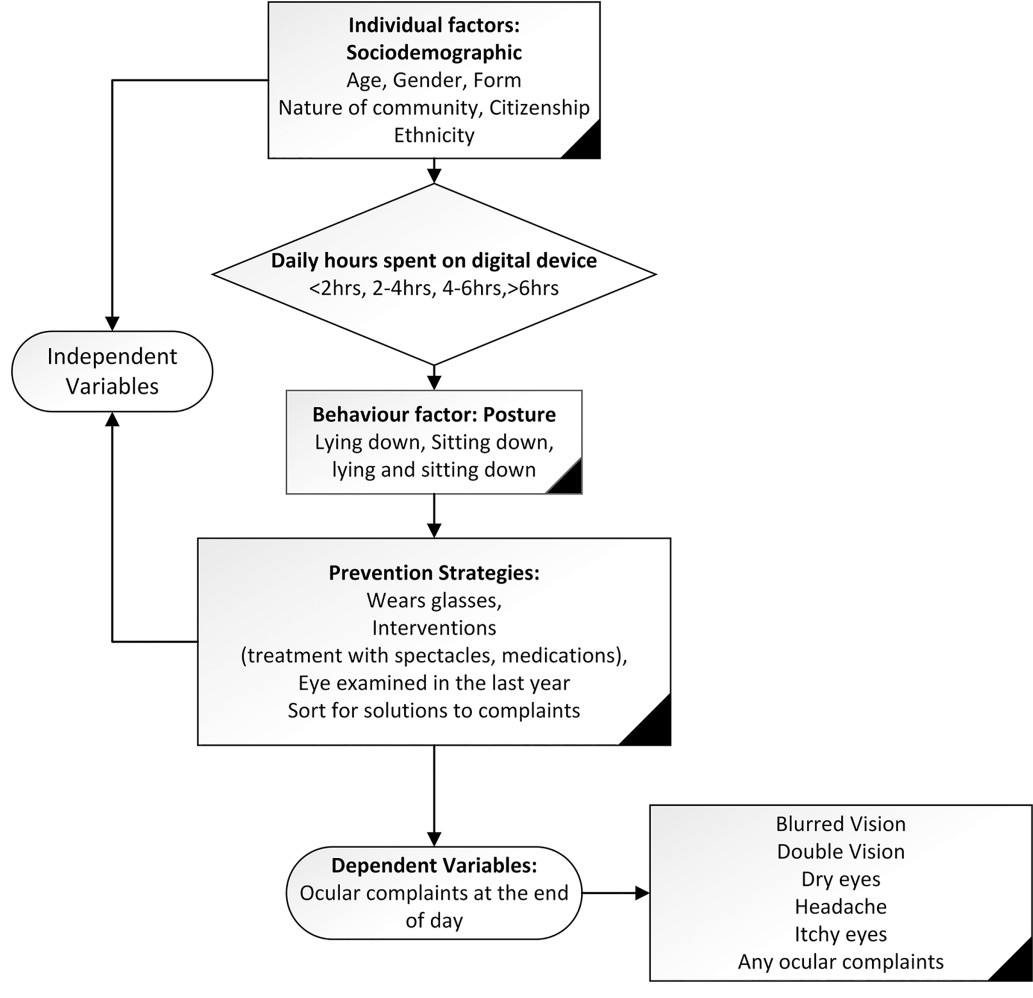

**Figure 1 The contextual framework showing the dependent and independent variables.**

## Independent variables

The independent variables were socio-demographic characteristics, average time spent per day learning remotely using a DD, the posture adopted while working on DD, prevention strategies to reduce ocular complaints. These were based on a previous study (*Ichhpujani et al., 2019*; *Mohan et al., 2021*), and details of these variables are shown in Fig. 1 which shows the contextual framework showing the dependent and independent variables.

## Data analysis

The data were analyzed using STATA version 14.1 (StataCorp. 2015. Release 14. College Station, TX, USA). To determine the level of ocular complaints among schoolchildren aged 12–19 years remotely using DD during the COVID-19 lockdown, the dependent variable was expressed as a dichotomous variable: category '0' if no ocular complaints and category '1' if ocular complaints. Firstly, the survey 'Svy' command was used to adjust the school and class sampling design to report frequency and percentage. The Taylor series

linearization method was used in the surveys to estimate confidence intervals around prevalence estimates. Cross-tabulations were generated to describe ocular complaints' frequencies and confidence intervals across independent variables. The statistical significances were tested using a chi-squared test. A two-level mixed-effect logistic regression analysis followed this. Level one represents the individual (class (form) characteristics) whereas level two is the cluster (school characteristics), and bivariate and multivariate multilevel binary logistic regression analyses were used to examine factors associated with ocular complaints. A stage modelling technique was used to examine the multivariate multilevel logistic regression model to account for school and class (form) variability. In the first stage, socio-demographic factors were entered into the baseline multiple regression model to examine factors associated with ocular complaints among the schoolchildren. A manual elimination method was conducted. Only significant variables were retained in the model (Model 1). In the second stage, average daily hours spent on a digital device were entered into model 1. Those factors with $p$-values $< 0.05$ were retained (model 2) after an elimination process. In the third stage, behavioral factors (posture) consisting of lying down, sitting down or both postures, when learning with the DD were added to model 2. As before, those factors with $p$-values $< 0.05$ were retained (model 3). Finally, a similar process was used for prevention strategies, which were added to model 3 and those factors with $p$-values $< 0.05$ were retained in the final model (model 4). Only those variables which were statistically associated with ocular complaints among the schoolchildren ($p < 0.05$) remained in the final model, and the unadjusted and adjusted odds ratios from a logistic model are presented with 95% confidence intervals (CI).

## RESULTS

### Characteristics of the participants

A total of 435 schoolchildren from seven secondary schools across T&T, the majority were citizens of T&T ($n = 421$, 96.8%) with an Indian ancestry ($n = 214$, 49.2%) and the minority, citizens of other countries ($n = 14$, 3.2%), participated in the study. Table 1 presents the characteristics of the participants. About half of the respondents (54.3%, $n = 236$) were in junior classes (forms 1–3), girls 50.6% ($n = 220$), and their ages ranged from 12 to 19 years, with 45% of the students aged 14–15 years.

Figure 2 presents the prevalence and 95% CI of the ocular complaints reported by the secondary school children who used DD for remote online learning during the COVID-19 pandemic lockdown. A total of 77.2% ($n = 335$) of the schoolchildren reported at least one ocular complaint, predominantly headache (75.00% 95% CI [71.17–78.47]) and blurred vision (65.10%, 95% CI [61.36–68.63]) at the end of day (Fig. 2). About one in three students (33.46%, 95% CI [27.43–40.09]) experienced double vision after using DD for learning at the end of the school day. Majority of those that experienced ocular complaints took various actions to resolve the complaints (274, 81.8%) mostly taking frequent breaks (60.3%, $n = 202$), while four persons did nothing (1.2%) and the rest did not respond to the question. A breakdown of the various actions taken by the school children to resolve the complaints are shown in Fig. 3.

**Table 1 Characteristics of the study sample including treatment type and frequency of digital device use.**

| Variables | Frequency, $n$ (%) |
|---|---|
| Socio-demographic factors | |
| Age category, years | |
| 12–13 | 70 (16.1) |
| 14–15 | 196 (45.1) |
| 16–17 | 111 (25.5) |
| 18–19 | 58 (13.3) |
| Gender | |
| Boys | 206 (47.4) |
| Girls | 220 (50.6) |
| Unspecified | 9 (2.1) |
| Nature of community | |
| Rural | 179 (41.2) |
| Urban | 256 (58.8) |
| Ethnicity | |
| Afro-Trinidad | 73 (16.8) |
| Indo-Trinidad | 214 (49.2) |
| Mixed | 108 (24.8) |
| Others | 40 (9.2) |
| Class Level | |
| Form 1–3 | 236 (54.3) |
| Form 4–6 | 199 (45.7) |
| Average daily hours spent on a digital device | |
| <2 h | 36 (8.3) |
| 2–4 h | 37 (8.5) |
| 4–6 h | 257 (59.1) |
| >6 h | 105 (24.1) |
| Behavioural factor: Posture | |
| *Laying down* | |
| No | 380 (87.4) |
| Yes | 55 (12.6) |
| *Sitting down* | |
| No | 119 (27.4) |
| Yes | 316 (72.6) |
| *Sitting & Laying down* | |
| No | 389 (89.4) |
| Yes | 46 (10.6) |
| Prevention strategies | |
| *Wore glasses when using a digital device* | |
| No | 286 (65.8) |
| Yes | 149 (34.2) |

(Continued)

| Table 1 (continued) | |
|---|---|
| **Variables** | **Frequency, *n* (%)** |
| Interventions | |
| *Treatment type* | |
| Spectacles≠ | 157 (36.1) |
| Medication | 49 (11.3) |
| No treatment | 229 (52.6) |
| *Eye examined in the last year* | |
| No | 330 (75.9) |
| Yes | 105 (24.1) |
| *Took actions to resolve complaints*P | |
| No | 61 (18.2) |
| Yes | 274 (81.8) |

**Notes:**
≠ Those with updated spectacles or newly prescribed.
P *Denominator was* number of those with ocular complaints (*n* = 335).
*n*, Number of subjects; %, Percentage.

### Time spent during the remote online learning using a digital device

The number of hours schoolchildren spent during online learning was shown in Table 1. Most of the students (about 91%) exceeded the recommended daily hours (<2 h), with only 8.3% (*n* = 36) meeting this recommendation for digital device use during the lockdown. About 59.1% (*n* = 257) of the students reported spending between 4–6 h on their DD while learning.

### The posture adopted during the remote online learning with the digital devices

Most of the schoolchildren surveyed (72.6%) reported that they sat down when using the DD for online learning, compared with the few (12.6%) who reported lying down when using the DD. Further analysis revealed that most children who used laptops/MacBooks for online learning did so predominantly when sitting down (67.72%; 95% CI [62.14–72.84]) or when sitting and laying down (58.7%; 95% CI [42.47–73.23]). On the other hand, most of those who used a mobile phone (45.45%; 95% CI [42.47–73.23]) and 21.82% of those who used a tablet/iPad reported lying down when using their devices. Although a hundred and forty-nine schoolchildren (34.2%) wore glasses when learning with the DD, only seven students (1.6%) reported change in glass prescription after using these DD, and less than a quarter of the students had an eye test in the last one year prior to data collection (Table 1).

### Differentials of ocular complaints among schoolchildren aged 12–19 years

The results of prevalence and chi-square analysis of ocular complaints are presented for dry eyes, headache, and blurred vision in Table 2 and for double vision, itchy eye and any ocular complaints in Table 3. The tables show that the prevalence of any ocular complaint increased with increasing age, varied with gender, rurality, class level, treatment type and was dependent on whether the student sort for solutions after experiencing their

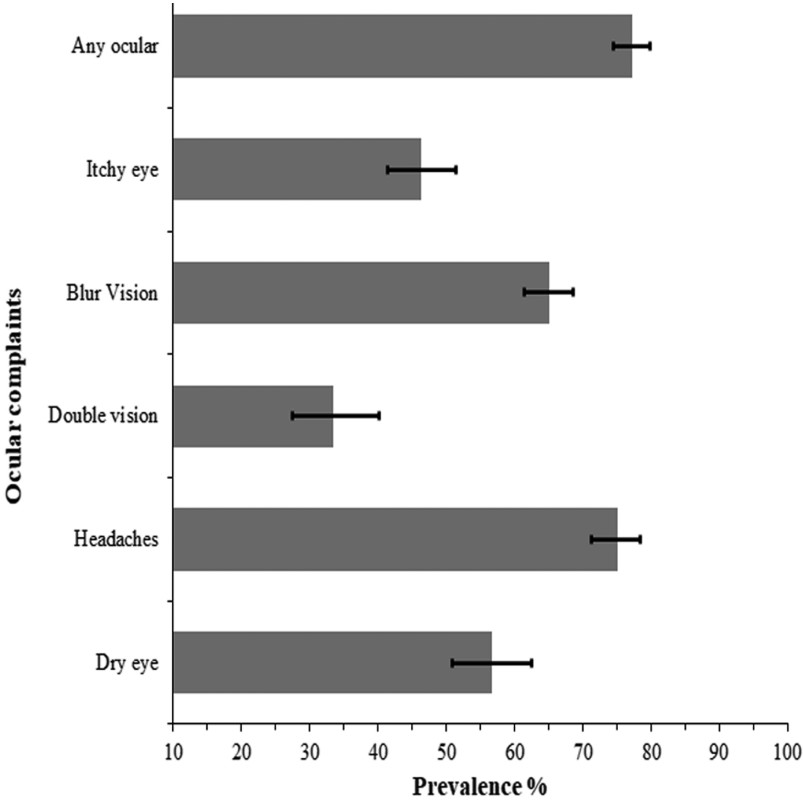

**Figure 2  Prevalence and 95% confidence intervals (CI) of ocular complaints from e-learning among school children during the lockdown in Trinidad and Tobago.**

symptoms. Females had a consistently higher prevalence of ocular complaints compared to males. Except for blurred vision (Table 2) and itchy eyes (Table 3), other ocular complaints were more prevalent in older students (18–19 years) than younger ones. Dry eye and blurred vision shown in Table 2 were more common among higher school graders, those who used spectacles and students who laid down when working with their devices for online learning.

### Factors associated with ocular complaints among students in T&T during the pandemic

The unadjusted analysis of factors associated with ocular complaints during online learning is presented as a Supplemental File (Tables S2 and S3). The results showed that age (13–19 years), gender (females), nature of residence (urban), class level (forms 4–6) were associated with any ocular complaints. However, after adjusting for the potential confounders in this study, older age (14–19 years), having an eye examination in the last year, taking actions to resolve the ocular complaints were associated with any ocular complaint following remote online learning using DD.

The adjusted odd ratios for factors associated with ocular complaints are presented in Tables 4 and 5 for dry eye, blurred vision, and headache and for double vision, itchy eye, and any ocular complaints, respectively. The table shows that increasing age was
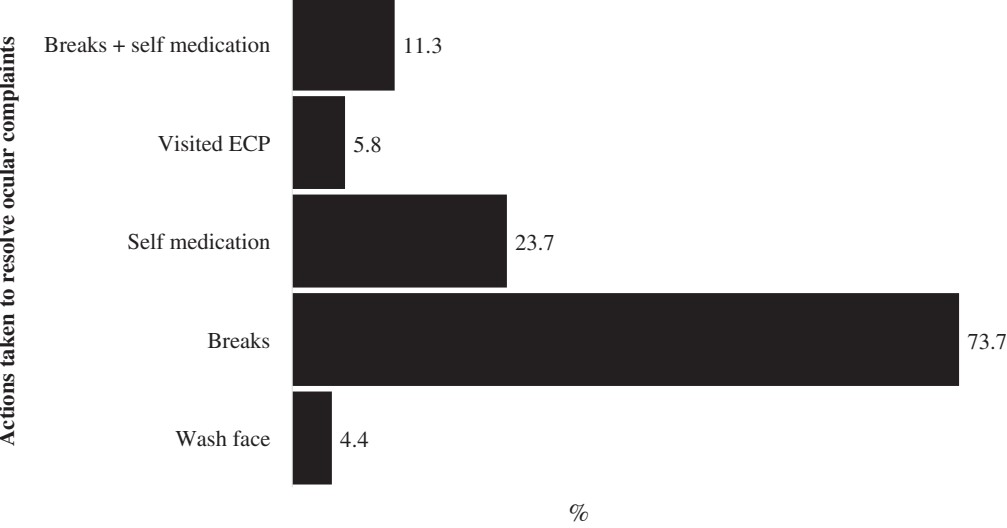

**Figure 3 Percentage distribution of actions taken by school children to resolve ocular complaints from e-learning during the lockdown in Trinidad and Tobago.**

significantly associated with dry eye, blurred vision, and headache. Being female was associated with blurred vision and headaches after adjustments. School children from other ethnic groups were less likely to report blurred vision than students from afro-Trinidad backgrounds, who used DD for learning during the lockdown. Posture, mainly lying down, increases the likelihood of itchy eyes at the end of the day. At the same time, the use of glasses when learning with a DD reduced the likelihood of blurred vision, headache but increased the likelihood of itchy eyes than not wearing glasses.

## DISCUSSION

This article examined prevalence and factors associated with ocular complaints among schoolchildren aged 12–19 years who used online e-learning during the COVID-19 Lockdown in T&T. The study found that three in four schoolchildren in T&T reported ocular complaints predominantly headache while one in three schoolchildren in this study reported double vision during the pandemic. The prevalence of ocular complaints increased with age and higher in older students with greater complaints of dry eye and blurred vision among higher school graders. Although many of the students were known spectacle lens wearers prior to the lockdown and adopted a sitting position when learning online using their devices, nearly all the students spent more than the recommended 2 h per day on their DD during the lockdown, and this was significantly associated with higher odds for ocular complaints in this study. The study also found older age to be associated with dry eye, blurred vision and headache, gender (female) to be associated with blurred vision and headaches while posture (lying down) and use of glasses for online learning was associated with itchy eyes.

In this study, headache and blurred vision were the most prevalent ocular complaints reported by schoolchildren in T&T who used DD for online e-learning during

Table 2 **Prevalence and 95% confidence intervals of dry eyes, headaches, and blurred vision ocular complaints during COVID-19 lockdown among schoolchildren age 12–19 years.**

| Variables | Dry eyes | F, *sig.* | Headache | F, *sig.* | Blurred vision | F, *sig.* |
|---|---|---|---|---|---|---|
| Socio-demographic factors | | | | | | |
| Age category, years | | | | | | |
| 12–13 | 42.37 [29.64–56.21] | 5.273, *p = 0.002* | 56.90 [44.46–68.52] | 5.284, *p = 0.002* | 47.06 [33.48–61.08] | 3.249, *p = 0.026* |
| 14–15 | 57.45 [49.85–64.70] | | 74.70 [66.61–81.38] | | 63.93 [54.48–72.42] | |
| 16–17 | 53.49 [43.57–63.14] | | 77.45 [69.52–83.80] | | 72.15 [61.72–80.64] | |
| 18–19 | 75.93 [62.29–85.76] | | 90.74 [76.46–96.73] | | 76.74 [60.06–87.87] | |
| Gender | | | | | | |
| Boys | 50.64 [42.65–58.60] | 4.321, *p = 0.043* | 67.43 [60.25–73.87] | 10.175, *p = 0.003* | 53.08 [45.61–60.41] | 15.108, *p = 0.0004* |
| Girls | 62.50 [53.65–70.59] | | 82.32 [76.33–87.06] | | 76.28 [69.43–82.00] | |
| Nature of community | | | | | | |
| Rural | 54.29 [46.03–62.31] | 0.702, *p = 0.406* | 72.05 [65.24–77.97] | 1.264, *p = 0.267* | 60.63 [53.81–67.06] | 2.640, *p = 0.111* |
| Urban | 58.50 [51.15–65.49] | | 77.17 [71.21–82.2] | | 68.45 [62.62–73.75] | |
| Ethnicity | | | | | | |
| Afro-Trinidad | 50.00 [38.31–61.69] | 0.802, *p = 0.480* | 75.38 [62.29–85.02] | 0.811, *p = 0.483* | 66.67 [52.74–78.19] | 2.029, *p = 0.116* |
| Indo-Tri | 60.98 [50.27–70.72] | | 78.65 [70.48–85.04] | | 68.35 [60.71–75.11] | |
| Mixed | 55.81 [43.32–67.61] | | 70.71 [60.82–78.96] | | 65.48 [53.49–75.77] | |
| Others | 50.00 [34.96–65.04] | | 68.42 [51.48–81.57] | | 38.10 [18.64–62.31] | |
| Year in school | | | | | | |
| Form 1–3 | 50.57 [43.80–57.32] | 6.527, *p = 0.014* | 70.05 [63.95–75.52] | 4.659, *p = 0.036* | 58.08 [50.54–65.23] | 4.563, *p = 0.038* |
| Form 4–6 | 63.41 [54.83–71.22] | | 80.33 [73.50–85.74] | | 72.86 [63.99–80.22] | |
| Average daily hours spent on digital device | | | | | | |
| <2 h | 34.62 [18.50–55.25] | 2.357, *p = 0.075* | 62.50 [43.30–78.44] | 1.954, *p = 0.129* | 57.69 [38.22–75.04] | 0.363, *p = 0.773* |
| 2–4 h | 50.00 [31.31–68.69] | | 65.52 [49.62–78.57] | | 64.00 [43.39–80.48] | |
| 4–6 h | 57.79 [50.44–64.81] | | 77.63 [72.56–81.99] | | 64.91 [59.85–69.66] | |
| >6 h | 62.37 [51.10–72.43] | | 76.00 [67.87–82.60] | | 68.49 [57.90–77.46] | |
| Behavioural factor: Posture | | | | | | |
| Lying down | | | | | | |
| No | 53.61 [47.11–59.99] | 7.468, *p = 0.009* | 73.86 [69.63–90.87] | 1.603, *p = 0.212* | 63.78 [59.78–67.60] | 1.556, *p = 0.219* |
| Yes | 75.51 [60.26–86.25] | | 82.35 [68.64–90.87] | | 73.17 [58.15–84.26] | |
| Sitting | | | | | | |
| No | 65.66 [55.26–74.74] | 4.218, *p = 0.046* | 79.61 [73.07–84.89] | 2.507, *p = 0.120* | 69.66 [61.27–76.92] | 1.711, *p = 0.197* |
| Yes | 53.11 [45.74–60.35] | | 73.29 [68.31–77.73] | | 63.11 [58.38–67.60] | |
| Sitting & Lying down | | | | | | |

(Continued)

| Table 2 (continued) | | | | | | |
|---|---|---|---|---|---|---|
| Variables | Dry eyes | F, *sig.* | Headache | F, *sig.* | Blurred vision | F, *sig.* |
| No | 57.65 [51.73–63.36] | 7.468, $p = 0.009$ | 74.19 [69.67–78.25] | 1.603, $p = 0.212$ | 64.26 [60.29–68.04] | 1.711, $p = 0.197$ |
| Yes | 48.48 [32.19–65.11] | | 82.05 [68.29–90.66] | | 71.88 [57.96–82.57] | |
| Prevention strategies | | | | | | |
| Wears glasses | | | | | | |
| No | 53.98 [47.25–60.57] | 2.491, $p = 0.122$ | 74.80 [69.79–79.34] | 0.0163, $p = 0.899$ | 65.08 [59.71–70.10] | 0.000, $p = 0.998$ |
| Yes | 62.28 [52.77–70.93] | | 75.38 [67.69–81.74] | | 65.09 [54.92–74.06] | |
| Interventions *Treatment type* | | | | | | |
| Spectacles | 66.96 [57.98–4.85] | 6.968, $p = 0.001$ | 83.82 [78.09–88.28] | 10.162, $p < 0.001$ | 72.48 [64.67–79.12] | 5.000, $p = 0.009$ |
| Medications | 71.43 [52.13–85.16] | | 84.62 [72.11–92.12] | | 75.00 [58.54–86.44] | |
| None | 47.89 [40.59–55.29] | | 67.32 [61.46–72. 68] | | 57.79 [52.56–62.86] | |
| *Eye exam in the last year* | | | | | | |
| No | 47.92 [41.51–54.41] | 32.356, $p < 0.001$ | 68.99 [64.77–72.91] | 35.490, $p < 0.001$ | 56.81 [52.19–61.31] | 37.036, $p < 0.001$ |
| Yes | 88.00 [77.40–94.01] | | 93.55 [87.18–96.87] | | 86.59 [79.32–91.57] | |
| *Took actions to resolve complaints* | | | | | | |
| No | 49.36 [42.60–56.14] | 14.113, $p < 0.001$ | 66.72 [61.00–71.15] | 18.219, $p < 0.001$ | 54.90 [50.29–59.43] | 19.653, $p < 0.001$ |
| Yes | 72.90 [61.97–81.62] | | 91.60 [83.13–96.02] | | 87.91 [76.50–94.20] | |

**Note:**
$p$–values are significant (sig.) from chi–square test at 5%, F values are shown and the degree of freedom df = 42 except for hours spent on digital devices.

COVID-19 pandemic lockdown. These findings are consistent with previous studies that reported a similar higher prevalence of headaches among school children who used DD (*Agarwal, Goel & Sharma, 2013*; *Dessie et al., 2018*; *Mohan et al., 2021*; *Portello et al., 2012*; *Shantakumari et al., 2014*), and for those studies that reported lower rates of headache (ranging from 17.9–50.2%)(*Bahkir & Grandee, 2020*; *Ichhpujani et al., 2019*; *Vilela et al., 2015*), they were not conducted during the lockdown period. However, a study in India found that most of the participants experienced heaviness (79.7%) and redness of the eye (69.1%) (*Gupta, Chauhan & Varshney, 2021*). In another study that determined the relationship between screen time (ST) and dry eye, researchers reported that eye fatigue was more prevalent than other ocular complaints in a pediatric population (*Elhusseiny et al., 2021*). While describing the game-based VDT activity in contrast to work-based VDT users, another study elucidated that ocular fatigue experienced by the users can be attributed to the high accommodation and convergence values required to maintain clear vision at near to VDTs throughout the game session (*Lee et al., 2019*).

The reported surge in the proportion of children using DD for more than the recommended 2 h a day during the lockdown (*Mohan et al., 2021*), was associated with an increase in the likelihood for reporting ocular complaints in this study. Considering that uncorrected refractive error or not using the prescribed correction has been linked to

**Table 3 Prevalence and 95% confidence intervals of double vision, itchy eyes and any ocular complaints during COVID-19 lockdown among schoolchildren age 12–19 years.**

| Variables | Double vision | F, *sig.* | Itchy eyes | F, *sig.* | Any ocular[#] | F, *sig.* |
|---|---|---|---|---|---|---|
| Socio-demographic factors | | | | | | |
| Age category, years | | | | | | |
| 12–13 | 30.00 [17.57–46.28] | 1.133, *p = 0.336* | 45.45 [31.06–60.65] | 1.682, *p = 0.176* | 62.86 [50.46–73.76] | 5.384, *p = 0.002* |
| 14–15 | 32.11 [24.15–41.26] | | 43.05 [35.86–50.54] | | 76.02 [70.07–81.10] | |
| 16–17 | 32.05 [22.55–43.31] | | 44.44 [33.85–55.57] | | 79.28 [69.77–86.38] | |
| 18–19 | 48.15 [31.22–65.52] | | 61.22 [48.41–2.65] | | 94.83 [84.87–98.36] | |
| Gender | | | | | | |
| Boys | 25.83 [18.50–34.83] | 4.719, *p = 0.0355* | 40.65 [33.53–48.18] | 5.860, *p = 0.020* | 72.33 [66.94–77.14] | 5.255, *p = 0.027* |
| Girls | 41.09 [30.94–52.04] | | 52.20 [45.66–58.66] | | 82.27 [76.45–86.90] | |
| Nature of community | | | | | | |
| Rural | 27.78 [20.82–36.01] | 5.124, *p = 0.028* | 42.86 [34.78–51.34] | 1.150, *p = 0.288* | 72.07 [66.14–77.31] | 6.549, *p = 0.014* |
| Urban | 37.67 [30.32–45.64] | | 48.78 [42.05–55.56] | | 80.86 [77.08–84.15] | |
| Ethnicity | | | | | | |
| Afro-Trinidad | 33.33 [21.05–48.39] | 0.364, *p = 0.774* | 39.62 [27.54–53.12] | 0.655, *p = 0.573* | 76.71 [66.28–84.66] | 1.532, *p = 0.212* |
| Indo-Tri | 35.14 [25.29–45.14] | | 49.1 [40.67–57.59] | | 80.84 [74.47–85.92] | |
| Mixed | 34.29 [23.57–46.88] | | 47.83 [37.29–58.56] | | 75.00 [65.34–82.68] | |
| Others | 24.00 [10.24–46.63] | | 39.39 [23.98–57.26] | | 65.00 [49.86–77.62] | |
| Year in school | | | | | | |
| Form 1–3 | 32.37 [25.41–40.21] | 0.169, *p = 0.683* | 43.48 [36.66–50.55] | 1.238, *p = 0.272* | 71.61 [66.66–76.09] | 8.759, *p = 0.005* |
| Form 4–6 | 34.78 [25.55–45.32] | | 49.69 [41.60–57.80] | | 83.92 [78.56–88.14] | |
| Average daily hours spent on DD | | | | | | |
| <2 h | 40.00 [24.10–58.33] | 1.784, *p = 0.160* | 32.00 [15.87–54.00] | 1.954, *p = 0.1292* | 69.44 [52.58–82.33] | 1.261, *p = 0.291* |
| 2–4 h | 18.18 [6.58–41.20] | | 38.46 [23.27–56.29] | | 67.57 [50.35–81.06] | |
| 4–6 h | 30.43 [22.36–39.93] | | 47.26 [40.96–53.65] | | 78.60 [74.79–81.97] | |
| >6 h | 42.03 [29.84–55.28] | | 50.54 [39.39–61.63] | | 80.00 [70.48–87.01] | |
| Behavioural factor: Posture | | | | | | |
| Lying down | | | | | | |
| No | 30.36 [24.85–36.50] | 8.426, *p = 0.005* | 43.42 [38.23–48.76] | 11.113, *p = 0.001* | 76.58 [73.24–79.62] | 0.817, *p = 0.371* |
| Yes | 56.67 [36.88–74.53] | | 68.29 [54.09–79.74] | | 81.82 [70.31–89.53] | |
| Sitting | | | | | | |
| No | 45.45 [32.83–58.69] | 6.283, *p = 0.016* | 54.35 [44.27–64.08] | 3.162, *p = 0.082* | 79.83 [73.15–85.19] | 0.711, *p = 0.403* |
| Yes | 29.26 [23.29–36.03] | | 43.48 [37.40–49.76] | | 76.27 [72.09–79.99] | |
| Sitting & Lying down | | | | | | |
| No | 33.78 [27.19–41.07] | 8.426, *p = 0.005* | 46.93 [41.36–52.57] | 11.113, *p = 0.001* | 76.61 [73.32–79.60] | 0.817, *p = 0.371* |
| Yes | 31.03 [16.90–49.90] | | 41.67 [27.78–57.02] | | 82.61 [70.18–90.56] | |
| Prevention strategies | | | | | | |

*(Continued)*

| Table 3 (continued) | | | | | | |
|---|---|---|---|---|---|---|
| Variables | Double vision | F, sig. | Itchy eyes | F, sig. | Any ocular[#] | F, sig. |
| Wears glasses | | | | | | |
| No | 33.92 [26.88–41.75] | 0.053, p = 0.818 | 45.33 [39.32–51.48] | 0.234, p = 0631 | 76.57 [72.78–79.98] | 0.817, p = 0.370 |
| Yes | 32.53 [23.14–43.58] | | 48.33 [38.28–58.52] | | 78.52 [71.17–84.41] | |
| Interventions Treatment type | | | | | | |
| Spectacles | 40.7 [30.94–51.25] | 3.132, p = 0.052 | 52.94 [43.83–61.86] | 7.648, p = 0.001 | 84.71 [79.34–88.88] | 5.049, p = 0.009 |
| Medications | 43.48 [23.76–65.50] | | 72.97 [51.61–87.24] | | 77.55 [64.17–86.95] | |
| None | 27.59 [21.39–34.78] | | 37.04 [30.55–44.02] | | 72.05 [67.43–76.25] | |
| Eye exam in the last year | | | | | | |
| No | 28.99 [23.05–35.74] | 9.078, p = 0.004 | 40.15[34.87–45.66] | 21.518, p = 0.000 | 72.42 [69.35–75.3] | 26.728, p < 0.001 |
| Yes | 53.19 [37. 15–68.60] | | 68.42 [57.22–77.83] | | 92.38 [86.44–95.85] | |
| Took actions to resolve complaints | | | | | | |
| No | 25.00 [19.24–31.81] | 26.053, p < 0.001 | 39.66 [34.90–44.61] | 14.41, p < 0.001 | 69.15 [65.09–72.94] | 32.669, p < 0.001 |
| Yes | 52.56 [41.39–63.48] | | 60.18 [49.63–69.85] | | 94.29 [88.88–97.15] | |

**Notes:**
[#] Any ocular symptom.
p–values are significant (sig.) from chi square test at 5%, F values are shown and the degree of freedom df = 42.

ocular complaints of headaches among children who used DD (*Gupta, Chauhan & Varshney, 2021*; *Portello et al., 2012*; *Vilela et al., 2015*), the higher prevalence of ocular complaints in this study may be related to the fact that majority of the schoolchildren either did not use spectacles when working with their devices, and/or not had an eye examination in the last year.

The schoolchildren who had a recent eye examination were more likely to report higher odds for all ocular complaints and those who took actions to resolve their complaints also reported higher odds for nearly all the ocular complaints. Of those who reported ocular complaints, majority took breaks from their devices, some used medications in order to resolve the complaints. Although we did not inquire on the length of the breaks and severity of their ocular complaints, the fact that many of them tried to resolve the complaints suggest that the complaints were concerning to them, which is in line with the high prevalence of ocular complaints particularly headaches in our study. The low uptake of eye examinations suggests that the school children in T&T may be unaware and may not express or understand the symptoms of uncorrected refractive error such as pain, headache and discomfort (*Gupta, Chauhan & Varshney, 2021*), which could remain undiagnosed and uncorrected in the absence of a comprehensive eye examination.

In the present study, we found that schoolchildren who were lying down when using their DD for online learning were significantly more likely to report itchy eyes than those schoolchildren not lying down. This finding was supported by a previous study that indicated that prolonged computer related task or use, were linked to ocular complaints including itchy eyes (*Sheppard & Wolffsohn, 2018*) and additional analysis using this

**Table 4 Factors associated with ocular complaints during COVID-19 lockdown among. Results are adjusted odds ratios (confidence intervals) from hierarchical multivariate analysis. Only the significant variables are presented.**

| Variables | Dry eyes | *p*-value | Blurred vision | *p*-value | Headache | *p*-value |
|---|---|---|---|---|---|---|
| Socio-demographic factors | | | | | | |
| Age category, years | | | | | | |
| 12–13 | 1.00 | | 1.00 | | 1.00 | |
| 14–15 | **2.22 [1.07–4.61]** | **0.032** | **3.15 [1.39–7.14]** | **0.006** | **3.23 [1.54–6.78]** | 0.002 |
| 16–17 | 1.47 [0.67–3.25] | **0.335** | **3.77 [1.49–9.52]** | **0.005** | **3.39 [1.47–7.83]** | 0.004 |
| 18–19 | **3.92 [1.50–10.20]** | **0.005** | **3.74 [1.29–10.80]** | **0.015** | **8.95 [2.75–29.20]** | 0.000 |
| Gender | | | | | | |
| Boys | – | | 1.00 | | 1.00 | |
| Girls | – | | **2.92 [1.60–5.32]** | **0.000** | **2.07 [1.19–3.58]** | 0.001 |
| Ethnicity | | | | | | |
| Afro-Trinidad | – | | 1.00 | | – | |
| Others | – | | **0.15 [0.04–0.57]** | 0.005 | – | |
| Behavioural factor: Posture | | | | | | |
| *Sitting* | | | | | | |
| No | 1.00 | | – | | – | |
| Yes | **0.39 [0.19–0.80]** | **0.010** | – | | – | |
| *Sitting & Laying down* | | | | | | |
| No | 1.00 | | – | | – | |
| Yes | **0.34 [0.13–0.93]** | 0.035 | – | | – | |
| Prevention strategies | | | | | | |
| *Wore glasses when using a digital device* | | | | | | |
| No | – | | 1.00 | | 1.00 | |
| Yes | – | | **0.48 [0.25–0.92]** | 0.026 | **0.47 [0.25–0.87]** | **0.016** |
| Interventions | | | | | | |
| *Eye exam in the last year* | | | | | | |
| No | 1.00 | | 1.00 | | 1.00 | |
| Yes | **7.64 [3.46–16.87]** | 0.000 | **7.92 [3.34–18.82]** | 0.000 | **8.30 [3.16–21.82]** | 0.000 |
| *Took actions to resolve ocular complaints* | | | | | | |
| No | 1.00 | | 1.00 | | 1.00 | |
| Yes | **2.88 [1.63–5.11]** | 0.000 | **5.61 [2.60–12.09]** | 0.000 | **5.33 [2.62–10.84]** | 0.000 |

**Note:**
Bolded confidence intervals are significant variables. Empty cells are non–significant variables in the unadjusted analysis.

data also revealed that schoolchildren who used mobile phones to do their schoolwork during COVID-19 lockdown significantly reported higher prevalence of lying down compared to both those who used a tablet or iPad (45.5% *vs.* 21.8) and children who used laptop or desktops about (32.7%) to do their schoolwork during COVID-19 while lying down.

With regards to posture adopted by schoolchildren during online learning using their DD, the study found that lying down was associated with a higher likelihood for ocular complaints of itchy eyes at the end of the day and this bad posture was more common among those who used mobile phones for e-learning during the lockdown. This finding is

**Table 5 Factors associated with ocular complaints during COVID-19 lockdown among. Results are adjusted odds ratios (confidence intervals) from hierarchical multivariate analysis. Only the significant variables are presented.**

| Variables | Double vision | p-value | Itchy eyes | p-value | Any ocular# | p-value |
|---|---|---|---|---|---|---|
| Socio-demographic factors | | | | | | |
| Age category, years | | | | | | |
| 12–13 | | | | | 1.00 | |
| 14–15 | – | | – | | **2.23 [1.16–4.27]** | **0.016** |
| 16–17 | – | | – | | **2.31 [1.09–4.89]** | **0.029** |
| 18–19 | – | | – | | **13.86 [3.62–52.97]** | **0.000** |
| Gender | | | | | | |
| Boys | – | | – | | – | – |
| Girls | – | | – | | – | – |
| Behavioural factor: Posture | | | | | | |
| *Lying down* | | | | | | |
| No | | | 1.00 | | – | – |
| Yes | | | **3.68 [1.77–7.68]** | 0.001 | – | – |
| *Sitting* | | | | | | |
| No | 1.00 | | – | | – | – |
| Yes | **0.28 [0.13–0.60]** | 0.001 | – | – | – | – |
| *Sitting & Laying down* | | | | | | |
| No | 1.00 | | – | | – | – |
| Yes | **0.25 [0.08–0.75]** | 0.014 | – | | – | – |
| Prevention strategies | | | | | | |
| *Wore glasses when using a digital device* | | | | | | |
| No | – | | 1.00 | – | 1.00 | |
| Yes | - | | **2.14 [1.32–3.49]** | **0.002** | **0.35 [0.16–0.76]** | **0.008** |
| *Treatment type* | | | | | | |
| Spectacles | – | | 1.00 | | 1.00 | |
| Medications | – | | **2.64 [1.12–6.21]** | 0.026 | 0.32 [0.11–0.93] | 0.036 |
| *Eye exam in the last year* | | | | | | |
| No | 1.00 | | 1.00 | | 1.00 | |
| Yes | **3.03 [1.50–6.14]** | 0.002 | **2.44 [1.28–4.67]** | 0.007 | **4.47 [1.86–10.75]** | 0.001 |
| *Took actions to resolve ocular complaints* | | | | | | |
| No | 1.00 | | – | – | 1.00 | |
| Yes | **3.45 [1.89–6.28]** | 0.000 | – | – | **7.94 [3.63–17.41]** | 0.000 |

Notes:
# Any ocular symptom.
Bolded confidence intervals are significant variables. Empty cells are non–significant variables in the unadjusted analysis.

supported by another study which reported that inappropriate sitting positions when using DDs increased the odds for ocular complaints by about 2.3 times when compared to appropriate sitting posture (*Assefa et al., 2017*). When people assume bad posture when using a DD, their eyes work harder to concentrate, and this causes the eye muscles to become more hyperactive and leads to increase in ocular complaints. These findings highlight the importance of regular eye examinations the need for an increase in the educational campaign among schoolchildren regarding the ocular impacts of online

learning using mobile phone, which is essential to prevent eye discomfort and stress that occur from poor posture.

Consistent with other studies (*Mohan et al., 2021*; *Sheppard & Wolffsohn, 2018*), this study found that older students were more likely to report dry eye, blurred vision and headache. This was not an unexpected finding since older children are in higher classes, and should be more engaged with online classes, and more likely to spend more time on their devices. The findings of higher prevalence of ocular complaints among girls than boys is supported by past studies (*Bahkir & Grandee, 2020*; *Mohan et al., 2021*; *Portello et al., 2012*); however, a higher prevalence of ocular complaints in males (*Ganne et al., 2020*) or similar prevalence between girls and boys (*Agarwal, Goel & Sharma, 2013*) have also been reported. It was suggested that the ability to multitask on DD increases the likelihood of ocular complaints in one gender than the other (*Mohan et al., 2021*). For females, their hormonal changes may alter the eye's homeostasis, which may cause blurred vision (*Gong et al., 2015*).

Regarding double vision among digital device users, data on the risk factors are lacking, even though double vision may indicate a more severe problem (convergence insufficiency) (*Alvarez et al., 2021*) aggravated during intense close work (*Barnhardt et al., 2012*). Studies have demonstrated a significant association between double vision and both short term (less than 4 h a day (*Nunes et al., 2018*)) and long term (more than 6 h a day (*Comério et al., 2017*)) use of the computer. The finding that the majority of the schoolchildren with double vision have had an eye examination prior to participating in the study suggests that the ocular complaints may have been resolved as the children may not have coped with the visual demands if the complaints of double vision were unresolved.

## STRENGTHS AND LIMITATIONS

There are some limitations of this study that should be considered when interpreting the findings. First, the study used self-reported data obtained *via* internet surveys making it difficult to verify the schoolchildren's responses. It is possible that students may deliberately exaggerate their symptoms for any reason, which cannot be objectively verified. Second, the study did not enquire about other ocular complaints such as foreign body sensation, watering and red eyes which make up the digital eye strain (*Sheppard & Wolffsohn, 2018*). An assessment of the severity of these symptoms among school children is needed in future studies. Third, this is a cross-sectional study and causal relationship cannot be established. Fourth, the study's age group restriction limits the generalization of our findings to schoolchildren in the age bracket. Fifth, the exclusion of students with ocular diseases other than nearsightedness, farsightedness, and astigmatism suggests that we may have missed some of these ocular diseases which may be risk factors of ocular complaints. However, the robust analysis used in this study was to nullify any effect their existing ocular conditions such as dry eye, allergic conjunctivitis, might have on the outcome measures. Lastly, the effect of ambient illumination shown to cause fatigue for the pupil's reflex in a previous study (*Mylona et al., 2020*) was not considered in this study, because its impact on ocular symptoms remain controversial (*Lee et al., 2011*; *Lin et al.,*

*2009*). There is a need for additional research with a broader age group to investigate whether similar findings can be obtained. Despite these limitations, this study has some strengths. We used a larger sample size when compared with previous studies (*Bahkir & Grandee, 2020*; *Kim et al., 2017*; *Logaraj, Madhupriya & Hegde, 2014*; *Mowatt et al., 2018*; *Nunes et al., 2018*; *Portello et al., 2012*; *Usgaonkar, Shet Parkar & Shetty, 2021*), and the selection of the secondary schools in the region was random which makes our results more representative of the secondary school pupils in T&T, hence mitigating bias and expectation among participants. Further studies in T&T particularly randomized studies are needed to draw a substantial conclusion in this population and such studies will benefit from including other visual complaints such as foreign body sensation, watering and red eyes, which constitutes computer vision syndrome (CVS) (*Sheppard & Wolffsohn, 2018*).

## Implications of the findings

From this study findings, students and teachers should be informed about the importance of regular eye examinations and that vision examinations should be included in primary health care services. Also, the findings indicate the need for raised awareness among parents and teaching staff regarding ocular complaints in children who use DD and ways of maintaining eye hygiene including interruptions during a computer session. This follows the American Optometry Association recommendation of 15-min break every 2 h working on a DD. The factors identified in this study can be used for designing public health interventions by targeting the 'at-risk population' to minimize the associated ocular complaints observed among secondary school children, which reached significant levels after 2 h of remote learning. Children's vision systems are more fragile than adults, and symptoms develop when the task's visual demand exceeds the task's visual capability (*Kozeis, 2009*). School children should adopt a comfortable seating position using adjustable chairs while attending online sessions. Adopting awkward postures like laying down is more likely to exacerbate ocular complaints. The information is also useful to eye care practitioners in T&T who may be witnessing an increase in the proportion of patients presenting with ocular complaints and the possible reasons for this complaint.

## CONCLUSIONS

The study sheds light on a critical modern eye health issue that compromises children's vision. During the COVID-19 lockdown, over three in four students in T&T reported any ocular complaints. Tailored interventional messages to reduce all forms of ocular complaints should target older students, particularly girls who regularly examine their eyes. Appropriate positioning should be encouraged among schoolchildren working online while ensuring proper ergonomic practice with the devices. Since the post-pandemic era will likely see many schools continue to provide remote learning or adopt a hybrid pattern, this can increase the prevalence of these visual problems further and lead to more severe convergence issues. Assessment of the severity of these symptoms among school children is needed in future studies. However, incorporating public health messages on lifestyle changes during the management and treatment of school children with ocular complaints is crucial, as well as educating parents, teachers, and

students on the need for an annual eye examination. Future studies, particularly in other Caribbean countries, are needed to assess the long-term effect of prolonged near work on children's vision and academic achievement following the pandemic.

# ACKNOWLEDGEMENTS

The authors would like to express their gratitude to all the parents, teachers, and principals of the secondary schools in Trinidad and Tobago where this study was done. We would also want to convey our heartfelt appreciation to our colleagues for their invaluable assistance and motivation throughout the research endeavor.

### Funding
The authors received no funding for this work.

### Competing Interests
The authors declare that they have no competing interests.

### Author Contributions
- Kingsley Ekemiri conceived and designed the experiments, performed the experiments, analyzed the data, prepared figures and/or tables, authored or reviewed drafts of the article, and approved the final draft.
- Ngozika Ezinne conceived and designed the experiments, analyzed the data, authored or reviewed drafts of the article, and approved the final draft.
- Khadeejah Kamalodeen conceived and designed the experiments, performed the experiments, authored or reviewed drafts of the article, and approved the final draft.
- Keomi Pierre conceived and designed the experiments, performed the experiments, authored or reviewed drafts of the article, and approved the final draft.
- Brandon Lalla conceived and designed the experiments, performed the experiments, authored or reviewed drafts of the article, and approved the final draft.
- Onyekachukwu Amiebenomo conceived and designed the experiments, performed the experiments, analyzed the data, authored or reviewed drafts of the article, and approved the final draft.
- Diane van Staden analyzed the data, authored or reviewed drafts of the article, and approved the final draft.
- Ferial Zeried performed the experiments, analyzed the data, prepared figures and/or tables, authored or reviewed drafts of the article, and approved the final draft.
- Chioma Ekemiri conceived and designed the experiments, performed the experiments, analyzed the data, authored or reviewed drafts of the article, and approved the final draft.
- Kingsley E. Agho analyzed the data, prepared figures and/or tables, authored or reviewed drafts of the article, performed data analysis and interpretation of results, and approved the final draft.
- Uchechukwu Levi Osuagwu conceived and designed the experiments, performed the experiments, analyzed the data, prepared figures and/or tables, authored or reviewed drafts of the article, performed data analysis and interpretation of the results, and approved the final draft.

### Human Ethics

The following information was supplied relating to ethical approvals (*i.e.*, approving body and any reference numbers):

The University of the West Indies' Ethics Committee granted ethical approval to carry out this study within its facilities (Ethical Application Ref: CREC-SA.0712/01/202).

### Data Availability

The raw measurements are available in the Supplementary File.

### Supplemental Information

Supplemental information for this article can be found online at http://dx.doi.org/10.7717/peerj.13334#supplemental-information.

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
