# Peer review of "Online e-learning during the COVID-19 lockdown in Trinidad and Tobago: prevalence and associated factors with ocular complaints among schoolchildren aged 11–19 years"

_PeerJ, doi:10.7717/peerj.13334_

## Round 0.1 · original submission · Minor Revisions

Three experts have reviewed the manuscript, and they found some merit in the study. The authors must carefully addressed all these comments before the manuscript can be considered for publication.

Reviewer 1 ·

Basic reporting

no comment

Experimental design

no comment

Validity of the findings

no comment

Additional comments

I have reviewed the article and this article was aimed to determine the prevalence and factors associated with ocular complaints among school children aged 11-19 years during the COVID-19. Over 75% of students reported ocular complaints about online learning. Though the topic has been studied before, I still think this manuscript could add useful information to CVS in the Covid-19 pandemic.

I have a few questions or suggestions, and I suggest a minor review.

1. CVS has over 10 symptoms, so why did you choose only 5 ocular complaints?

2. Sample selection should be stated in detail. For example, how could you select seven schools from Trinidad's seven districts? Did students selected deliberately exaggerate the symptoms because they knew the aim of this study?

3. How could you distinguish the CVS symptom from symptoms from other ocular diseases? According to methods, students with other ocular diseases did not be excluded from the study.

4. The majority of participants were citizens of T&T (n=421, 96.8%), and it should be made clear the minority of them.

5. Another strength of this study was that the schoolchildren in this study were given adequate time to respond to the items voluntarily, and it is improbable that they completed the questions rapidly. The mean time students spent filling out questionnaires should be stated to prove this strength.

·

Basic reporting

No comment

Experimental design

No comment

Validity of the findings

No comment

Additional comments

This is an interesting study aimed at evaluating the prevalence and associated factors with ocular complaints among schoolchildren aged 11-19 years with online e-learning during the COVID-19 lockdown in Trinidad and Tobago. Thank you very much for the opportunity of reviewing this article that greatly adds to the scientific body of knowledge. I congratulate the authors for the work performed. Hereunder, a few minor comments that could help the readers in the understanding of the article and its background are presented:

In the abstract:
I recommend the authors to include:
- Number of subjects that responded to the questionnaire
- “Mean age of children was (include the mean age ± standard deviation here)”.
- Add “self-reported”, in Results: “The prevalence of self-reported headache …”

Strengths and Limitations section:
- It seems that ambient illumination was not considered. Please, include it as a limitation.

References:
There are two duplicate references:
- Lines 570-576: Portello, J. K., Rosenfield, M., Bababekova, Y. Estrada, J. M., & Leon, A. (2012). Please, leave only one of both.
Lines 595-600: Usgaonkar, U., Shet Parkar, S. R., & Shetty, A. (2021). Please, leave only one of both.

Minor changes:
- Lines 218-219, two lines that should be from the same paragraph are divided. Please, unify it.
- Lines 316, 333, 382. Remove the initial “J” before the citation “Portello”.
- In the title of Figure 2 it is written “Figure 3”.
- The results presented in Table 4 are not clear. Please, present it with “more order”.

Reviewer 3 ·

Basic reporting

The manuscript entitled "Online e-learning during the COVID-19 Lockdown in Trinidad and Tobago: Prevalence and Factors Associated with Ocular complaints among schoolchildren aged 11-19 years" aims to analyze the Computer Vision Syndrome -related symptoms during online learning in Trinidad and Tobago.
In my opinion, this manuscript is very interesting and it contributes to the knowledge of CVS.

Experimental design

Some comments:
- Since the data were collected for 4 months, do the authors think that the results may have any bias between participants? I understand that the time that participants spent at home significantly increase the symptoms.
- In my opinion, the manuscript is too long and some parts are unnecessary. Together with some minor spelling mistakes that need to be correct, it could be shortened (especially “data analysis”, and the tables, only the variables statistically significant could be shown and delete the rest of information).
- It would be interesting to include the questionnaire used for the data compilation as an appendix, because it is not clear.
- I understand that the authors only asked whether the participants suffered from these 5 symptoms. According to the definition of CVS, there are more ocular and visual symptoms related to the increased use of electronic devices. Moreover, it is difficult to analyze these symptoms without any information of the severity of them. It would be interesting for further studies to take this into account.

Validity of the findings

The results support the conclusion.

---

## Round 0.2 · Minor Revisions

Please consider the remaining comment of reviewer 1, and include it as a limitation of the article or justify your decision. After it, this paper can be considered for publication

Reviewer 1 ·

Basic reporting

no comment

Experimental design

no comment

Validity of the findings

no comment

Additional comments

I have reviewed the revised article and I think the authors answered questions and revised the manuscript seriously. Thus, I suggest an acceptance.

Though I suggest an acceptance, I still have an important suggestion. A minor review may be needed before acceptance.

The suggestion is as follows:

1. According to methods, students with ocular diseases other than nearsightedness, farsightedness, and astigmatism were excluded in this study, which is inappropriate from my view for these ocular diseases may be risk factors of ocular complaints. In addition, students with some ocular diseases, such as dry eye, allergic conjunctivitis, should be excluded from this study, because these diseases indeed have ocular symptoms and could be affected by seasons or other factors. This is a big limitation of this study, which should raise concerns of authors.

·

Basic reporting

No comment

Experimental design

No comment

Validity of the findings

No comment

Additional comments

The authors have performed a good job with the modifications performed in the revised version of the manuscript.
It is an interesting article that, in a simple way, brings knowledge to the subject treated. Congratulations to the authors for transmitting science and highlighting the importance of ocular complaints from digital devices for online learning.

---

## Round 0.3 · accepted · Accept

The authors have done a good job, and no further modifications are required.